# Reward-Guided Iterative Refinement in Diffusion Models at Test-Time with Applications to Protein and DNA Design

**Masatoshi Uehara** [* 1]  **Xingyu Su** [* 2]  **Yulai Zhao** [3]  **Xiner Li** [2]  **Aviv Regev** [1]
**Shuiwang Ji**[† 2]  **Sergey Levine**[† 4]  **Tommaso Biancalani**[† 1]

## Abstract

To fully leverage the capabilities of diffusion models, we are often interested in optimizing downstream reward functions during inference. While numerous algorithms for reward-guided generation have been recently proposed due to their significance, current approaches predominantly focus on single-shot generation, transitioning from fully noised to denoised states. We propose a novel framework for test-time reward optimization with diffusion models. Our approach employs an iterative refinement process consisting of two steps in each iteration: noising and reward-guided denoising. This sequential refinement allows for the gradual correction of errors introduced during reward optimization. Finally, we demonstrate its superior empirical performance in protein and cell-type specific regulatory DNA design. The code is available at https://github.com/masa-ue/ProDifEvo-Refinement.

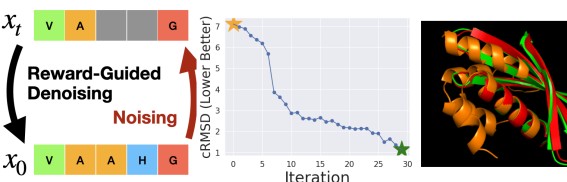

Figure 1: Our proposed framework follows an iterative process, with each iteration injecting noise into the sample and then denoising it while optimizing rewards. For sequences, this can be implemented via masked diffusion, initialized from pre-trained diffusion models (left). Our algorithm can continuously refine the outputs by gradually correcting errors introduced during reward-guided denoising, improving the design over successive iterations (middle). For instance, for the task of optimizing the similarity (RMSD) of a protein to a target structure (Red), we can progressively minimize the RMSD through refinement, optimizing the design from an initial (Orange) fit to a better final fit (Green), as shown on the right.

## 1. Introduction

Diffusion models have achieved significant success across various domains, including computer vision and scientific fields (Ramesh et al., 2021; Watson et al., 2023). These models enable sampling from complex natural image spaces or molecular spaces that resemble natural structures. Beyond the capabilities of such pre-trained diffusion models, there is often a need to optimize downstream reward functions. For instance, in text-to-image diffusion models, the reward function may be the alignment score (Black et al., 2023; Fan et al., 2023; Uehara et al., 2024), while in protein sequence diffusion models, it could include metrics such as stability, structural constraints, or binding affinity (Verkuil et al., 2022), and in DNA sequence diffusion models, it may involve activity levels (Sarkar et al., 2024; Lal et al., 2024).

Building on the motivation above, we focus on optimizing downstream reward functions while preserving the naturalness of the designs. (e.g., a natural-like protein sequence exhibiting strong binding affinity) by seamlessly integrating these reward functions with pre-trained diffusion models during inference. While numerous studies have proposed to incorporate rewards into the generation process of diffusion models (e.g., classifier guidance (Dhariwal and Nichol, 2021) by setting rewards as classifiers, derivative-free methods (Wu et al., 2024; Li et al., 2024)), they rely on a *single-shot* denoising pass for generation. However, a natural question arises:

*Can we further leverage inference-time computation during generation to refine the model's output?*

*Equal contribution [1]EvolutionaryScale (Work done at Genentech) [2]Texas A&M University [3]Princeton University [4]UC Berkley. Correspondence to: Masatoshi Uehara <masatoshi.uehara@evolutionaryscale.ai>, Xingyu Su <xingyu.su@tamu.edu>.

*Proceedings of the $42^{nd}$ International Conference on Machine Learning*, Vancouver, Canada. PMLR 267, 2025. Copyright 2025 by the author(s).

In this study, we observe that diffusion models can inherently support an *iterative* generation procedure, where the design can be progressively refined through successive cycles of masking and noise removal. This allows us to utilize arbitrarily large amounts of computation during generation to continuously improve the design.

Motivated by the above observations, we propose a novel framework for test-time reward optimization with diffusion models. Our approach employs an iterative refinement algorithm consisting of two steps in each iteration: partial noising and reward-guided denoising as in Figure 1. The reward-guided denoising step transitions from partially noised states to denoised states using techniques such as classifier guidance or derivative-free guidance. Unlike existing *single-shot* methods, our approach offers several advantages. First, our sequential refinement process allows for the gradual correction of errors introduced during reward-guided denoising, enabling us to optimize complex reward functions, such as structural properties in protein sequence design. In particular, this correction is expected to be crucial in recent successful masked diffusion models (Sahoo et al., 2024; Shi et al., 2024), as once a token is demasked, it remains unchanged until the end of the denoising step. Besides, for reward functions with hard constraints, commonly encountered in biological sequence or molecular design (e.g., cell-type-specific DNA design (Gosai et al., 2023; Lal et al., 2024) or binders with high specificity), our framework can effectively optimize such reward functions by initializing seed sequences within feasible regions that satisfy these constraints.

Our contribution is summarized as follows. First, we propose a new reward-guided generation framework for diffusion models that sequentially refines the generated outputs (Section 3). Our algorithm addresses two major issues in existing methods such as the lack of a correction mechanism and difficulties of handling hard constraints. Secondly, we provide a theoretical formulation demonstrating that our algorithm samples from the desirable distribution $\exp(r(x))p^{\mathrm{pre}}(\cdot)$, where $p^{\mathrm{pre}}(\cdot)$ is a pre-trained distribution (Section 4) and $r(\cdot)$ is a reward function. Finally, we present a specific instantiation of our unified framework by carefully designing the reward-guided denoising stage in each iteration, which bears similarities to evolutionary algorithms (Section 5). Using this approach, we experimentally demonstrate that our algorithm effectively optimizes reward functions, outperforming existing methods in computational protein and DNA design (Section 6).

### 1.1. Related Works

We categorize related works into three key aspects.

**Guidance (i.e. test-time reward optimization) in diffusion models.** Most classical approaches involve classifier guidance (Dhariwal and Nichol, 2021; Song et al., 2021), which adds the gradient of reward models (or classifiers) during inference. As reviewed in Uehara et al. (2025), recently, derivative-free methods such as SMC-based guidance (Wu et al., 2024; Dou and Song, 2024; Phillips et al., 2024; Cardoso et al., 2023) or value-based sampling (Li et al., 2024; 2025) have been proposed. However, these methods rely on single-shot generation from noisy states to denoised states. In contrast, we propose a novel iterative refinement approach that enables the optimization of complex reward functions, which can be challenging for single-shot reward-guided generation.

Note while classifier-free guidance (Ho and Salimans, 2022) and RL (control)-based fine-tuning (e.g., Fan et al. (2023); Black et al. (2023); Domingo-Enrich et al. (2024); Uehara et al. (2024); Venkatraman et al. (2024); Zhang et al. (2024)) also aim to address reward optimization in diffusion models, they are orthogonal to our work, as we focus on test-time techniques without any training.

**Refinement in language models.** Refinement-style generation has been explored in the context of BERT-style masked language models and general language models (Novak et al., 2016; Guu et al., 2018; Wang and Cho, 2019; Welleck et al., 2022; Padmakumar et al., 2023; Wang et al., 2024). However, our work is the first attempt to study iterative refinement in diffusion models. Note that while some readers may consider editing in diffusion models (e.g., Huang et al. (2024)) to be relevant , this is a distinct area, as the focus is not on reward optimization, unlike our work.

**Evolutionary algorithms and MCMC for biological sequence design.** Refinement-based approaches with reward models, such as variants of Gibbs sampling and genetic algorithms, have been widely used for protein/DNA design (Anishchenko et al., 2021; Jendrusch et al., 2021; Hie et al., 2022; Gosai et al., 2023; Pacesa et al., 2024). However, most works do not address the integration of diffusion models. While some studies focus on integrating generative models (Hie et al., 2024; Chen et al., 2024), we explore an approach tailored to diffusion models, given the recent success of diffusion models in protein and DNA sequence generation (Alamdari et al., 2023; Wang et al., 2024).

We finally remark that as closely related works, evolutionary-type algorithms with generative models have been actively investigated (Schneuing et al., 2024; Lee et al., 2025). Compared to these works, we use reward-guided decoding methods instead of standard decoding ways.

## 2. Preliminaries

We first provide an overview of diffusion models, then discuss current reward-guided algorithms in diffusion models and the potential challenges, which motivate our proposal.

### 2.1. Diffusion Models

In diffusion models, the objective is to learn a sampler $p^{\text{pre}}(\cdot) \in \Delta(\mathcal{X})$ for a given design space $\mathcal{X}$ using available data. The training procedure is summarized as follows. First, we define a forward noising process (also called a policy) $q_t : \mathcal{X} \to \Delta(\mathcal{X})$ that proceeds from $t = 0$ to $t = T$. Next, we learn a reverse denoising process $p_t : \mathcal{X} \to \Delta(\mathcal{X})$ parametrized by neural networks, ensuring that the marginal distributions induced by these forward and backward processes match.

To provide a concrete illustration, we explain masked diffusion models. However, we remark that our proposal in this paper can be applied to *any* diffusion model.

**Example 1** (Masked Diffusion Models)**.** *Here, we explain masked diffusion models (Sahoo et al., 2024; Shi et al., 2024; Austin et al., 2021; Campbell et al., 2022; Lou et al., 2023)).*

*Let $\mathcal{X}$ be a space of one-hot column vectors $\{x \in \{0,1\}^K : \sum_{i=1}^{K} x_i = 1\}$, and $\mathrm{Cat}(\pi)$ be the categorical distribution over $K$ classes with probabilities given by $\pi \in \Delta^K$ where $\Delta^K$ denotes the $K$-simplex. A typical choice of the forward noising process is $q_t(x_{t+1} \mid x_t) = \mathrm{Cat}(\alpha_t x_t + (1 - \alpha_t)\mathbf{m})$ where $\mathbf{m} = [\underbrace{0, \cdots, 0}_{K-1}, \mathrm{Mask}]$. Then, defining $\bar{\alpha}_t = \Pi_{i=1}^{t}\alpha_i$, the backward process is parameterized as*

$$x_{t-1} = \begin{cases} \delta(\cdot = x_t) & \text{if } x_t \neq \mathbf{m} \\ \mathrm{Cat}\left(\frac{(1-\bar{\alpha}_{t-1})\mathbf{m}+(\bar{\alpha}_{t-1}-\bar{\alpha}_t)\hat{x}_0(x_t;\theta)}{1-\bar{\alpha}_t}\right), & \text{if } x_t = \mathbf{m}, \end{cases}$$

*where $\hat{x}_0(x_t)$ is a predictor from $x_t$ to $x_0$.*

**Notation and remark.** $\delta_a$ denotes the Dirac delta distribution at mass $a$. With a slight abuse of notation, we express the initial distribution as $p_{T+1} : \mathcal{X} \to \Delta(\mathcal{X})$, and denote $[1, \cdots, T]$ by $[T]$.

### 2.2. Single-Shot Reward-Guided Generation

Our goal is to generate a natural-like design with a high reward. In particular, we focus on inference-time algorithms that do not require fine-tuning of pre-trained diffusion models. Below, we provide a summary of these methods.

For reward-guided generation, we often aim to sample from

$$p^{(\alpha)} := \underset{p \in \Delta(\mathcal{X})}{\mathrm{argmax}} \, \mathbb{E}_{x \sim p}[r(x)] - \alpha \mathrm{KL}(p\|p^{\text{pre}}) \quad (1)$$

$$= \exp(r(\cdot)/\alpha)p^{\text{pre}}(\cdot)/C,$$

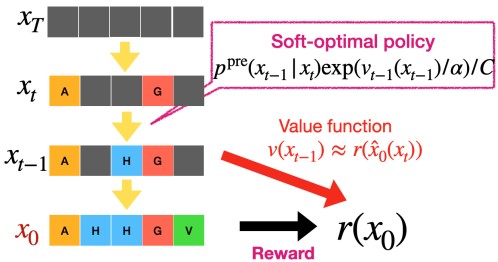

Figure 2: Existing reward-guided algorithms can be viewed as sequentially sampling from $x_T$ to $x_0$ following the soft optimal policy $\{p_t^\star\}_{t=T}^1$. The primary distinction among these algorithms lies in how $p_t^\star$ is approximated.

where $C$ is the normalizing constant. This objective is widely employed in generative models, such as RLHF in large language models (LLMs) (Ziegler et al., 2019; Ouyang et al., 2022). In diffusion models (e.g., Uehara et al. (2024, Theorem 1)), this is achieved by sequentially sampling from the *soft optimal policy* $\{p_t^\star\}_t$ from $t = T + 1$ to $t = 1$, which is defined by

$$p_t^\star(\cdot \mid x_t) \propto \exp(v_{t-1}(\cdot)/\alpha)p_t^{\text{pre}}(\cdot \mid x_t),$$

where

$$v_t(x_t) := \alpha \log \mathbb{E}_{x_0 \sim p^{\text{pre}}(x_0|x_t)}[\exp(r(x_0)/\alpha)|x_t]. \quad (2)$$

and the expectation is taken w.r.t. the pre-trained policy. Here, as illustrated in Figure 2, $v_{t-1}$ serves as a look-ahead function that predicts the reward at $x_0$ from $x_t$, often referred to as the *soft value function* in RL (or the optimal twisting proposal in SMC literature (Naesseth et al., 2019)).

In practice, we cannot precisely sample from soft optimal policies because (1) the soft value function $v_t$ is unknown, and (2) the action space under the optimal policy is large. Current algorithms address these challenges as follows.

**(1): Approximating soft value functions.** A typical approach is to use $r(\hat{x}_0(x_t))$ by leveraging the decoder $\hat{x}_0(x_t)$ obtained during pre-training. This approximation arises from replacing the expectation over $x_0 \sim p^{\text{pre}}(x_0|x_t)$ in (2) with $\delta_{\hat{x}_0(x_t)}$ (i.e., a Dirac delta at the mean of $p^{\text{pre}}(x_0|x_{t-1})$). Note its accuracy degrades as $t$ increases (i.e., as the state becomes more noisy). Despite its potential crudeness, this approximation is commonly adopted due to its training-free nature and the strong empirical performance demonstrated by methods such as DPS (Chung et al., 2022), reconstruction guidance (Ho et al., 2022), universal guidance (Bansal et al., 2023), and SVDD (Li et al., 2024).

**(2): Handling large action space.** Even with accurate value functions, sampling from the soft optimal policy still exhibits difficulty because its sample space $\mathcal{X}$ is still large.

Hence, we often resort to approximation techniques as follows.

- Classifier Guidance: In continuous diffusion models, the pre-trained policy $p_{t-1}^{\text{pre}}(\cdot \mid x_{t-1})$ is a Gaussian policy. By constructing *differentiable* value function models, we can approximate $p_t^\star$ by shifting the mean using $\nabla v_t(\cdot)/\alpha$. A similar approximation also applies to discrete diffusion models (Nisonoff et al., 2024).

- Derivative-Free Guidance: Another approach is using importance sampling (Li et al., 2024). Specifically, we generate several samples from $p_{t-1}^{\text{pre}}(\cdot \mid x_{t-1})$ and then select the next sample based on the importance weight $\exp(v_t(\cdot)/\alpha)$. A closely related method using Sequential Monte Carlo (SMC) has also been proposed, as discussed in Section 1.1.

### 2.3. Challenges of Single-Shot Generation

There are two main challenges with the aforementioned current algorithms. First, for certain complex reward functions, they may fail to fully optimize the rewards. This occurs because the value functions employed in these algorithms have approximation errors. When a value function model is inaccurate, the decision at that step can be suboptimal, and there is no correction mechanism during generation. This issue can be particularly severe in recent popular masked discrete diffusion models in Example 1, where once a token changes from the masking state, it remains unchanged until the terminal step ($t = 0$) (Sahoo et al., 2024; Shi et al., 2024). Consequently, any suboptimal token generation at intermediate steps cannot be rectified.

Another related challenge lies in accommodating hard constraints with a set $\mathcal{C} \subset \mathcal{X}$. Although one might assume that simply setting $r(\cdot) = I(\cdot \in \mathcal{C})$ would suffice, in practice, the generated outputs often fail to meet these constraints. This difficulty again arises from the inaccuracy of value function models at large $t$ (i.e., in highly noised states).

## 3. Iterative Refinement in Diffusion Models

To tackle challenges discussed in Section 2.3, we propose a new iterative inference-time framework for reward optimization in diffusion models. Our algorithm is an iterative algorithm where each step consists of two procedures: noising using forward pre-trained policies and reward-guided denoising using soft optimal policies. This framework is formalized in Algorithm 1.

Compared to existing algorithms that only perform single-shot denoising from $t = T$ to $t = 0$, our algorithm repeatedly performs reward optimization, as depicted in Figure 3. The challenge of single-shot algorithms – namely, the lack of a correction mechanism discussed in Section 2.3 – can

---

**Algorithm 1** Reward-Guided Evolutionary Refinement in Diffusion models (**RERD**)

1: **Require**: initial designs $x_0^{\langle 0 \rangle}$ (the index $\langle \cdot \rangle$ means the number of iteration steps), noise level $K$
2: **for** $s \in [0, \cdots, S-1]$ **do**
3:    *Noising*: Sample $x_K^{\langle s+1 \rangle}$ from $q_K(\cdot \mid x_0^{\langle s \rangle})$ where $q_K$ is a noising policy from $x_0$ to $x_K$ (See Section 2.1).
4:    *Reward-Guided Generation*: Sequentially sample from $\{p_t^\star\}_{t=K}^1$ (i.e., from $x_K^{\langle s+1 \rangle}$ to $x_0^{\langle s+1 \rangle}$) (In practice, we need to *approximate* it. Refer to Algorithm 2).
5: **end for**
6: **Output**: $\{x_0^{\langle S \rangle}\}$

---

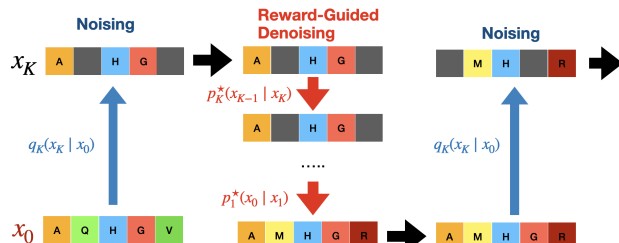

Figure 3: Summary of **RERD**: We instantiate it within masked diffusion models. It alternates reward-guided denoising and noising.

be addressed in **RERD**, by sequentially refining the outputs.

In Algorithm 1, several choices are important, which are outlined below.

- **Initial designs** $x_0^{\langle 0 \rangle}$: Here, we consider two approaches. The first choice is to run $\{p_t^\star\}$ from $t = T$ to $t = 0$ as in single-shot inference-time alignment algorithms. Second, if we have access to real data $\{z^i\} \sim p^{\text{pre}}(\cdot)$, we select samples with high rewards as initial designs. A straightforward way is by using the weighted empirical distribution:

$$\sum_i \frac{\exp(z^i/\alpha)}{\sum_j \exp(z^j/\alpha)} \delta_{z^i}. \qquad (3)$$

- **Approximation of the soft optimal policy** $p_t^\star$ **in Line 4**: As mentioned in Section 2.2, exact sampling from $p_t^\star$ is infeasible. However, we can employ any off-the-shelf methods to approximate it, such as classifier guidance or IS-based approaches discussed in Section 2.2. A specific instantiation of this approximation is considered in Section 5.

- **Noise level** $K$: When $K$ is close to $0$, the inference time per loop is reduced. Moreover, because value

function models used to approximate soft-optimal policies are typically more precise around $K = 0$ (see Section 2.2), the reward optimization step becomes more effective. On the other hand, using a larger $K$ allows for more substantial changes in a single step. In practice, striking the balance, we recommend setting $K/T$ low.

Next, we provide theoretical clarifications of our framework in Section 4. Additionally, we present a practical instantiation of our framework in Section 5.

## 4. Theoretical Analysis

We present the theoretical analysis of **RERD**. We begin with the key theorem, which clarifies its target distribution.

**Theorem 1** (Target Distribution of **RERD**). *Suppose (a) the initial design $x_0^{\langle 0 \rangle}$ follows $p^{(\alpha)}$ (defined in (1)), (b) the marginal distributions induced by the forward noising process match those of the learned noising process in the pre-trained diffusion models. Then, the output $x_0^{\langle S \rangle}$ from RERD follows the target distribution*

$$p^{(\alpha)}(\cdot) \propto \exp(r(\cdot)/\alpha)p^{\mathrm{pre}}(\cdot).$$

First, we discuss the validity of the assumptions. The assumption (a) is readily satisfied when using the introduced strategy of initial designs in Section 3. The assumption (b) is also mild, as pre-trained diffusion models are trained in this manner (Song et al., 2021), though certain errors may arise in practice. Another implicit assumption in practice is that we can approximate soft-optimal policies accurately.

Next, we explore the implications of Theorem 1. The central takeaway is that we can sample from a desired distribution for our task $p^{(\alpha)}$ in (1). Although this guarantee appears to mirror existing single-shot algorithms discussed in Section 2.2, we anticipate differing practical performance in terms of rewards. This is due to their robustness against errors in soft value function approximation $v_t(x_t) \approx r(\hat{x}_0(x_t))$.

To clarify, recall that in reward-guided algorithms, we must employ *approximated* soft value function models when sampling from the soft optimal policies $p_t^\star \propto \exp(v_{t-1}(\cdot)/\alpha)p_{t-1}^{\mathrm{pre}}(\cdot \mid x_t)$. The approximation often becomes more precise as the time step $t$ in the soft optimal policy approaches 0, as mentioned in Section 2.2. Indeed, in the extreme case, when $t = 0$, the exact equality holds. Therefore, by maintaining a sufficiently small noise level $t = K$ and avoiding the approximation of value functions at large $t$, **RERD** can effectively minimize approximation errors in practice.

**Sketch of the Proof of Theorem 1.** The detailed proof is deferred to Section A. In brief, first, we

show that the marginal distribution after noising is $p_K^{\mathrm{pre}}(\cdot)\exp(v_K(\cdot)/\alpha)/C$ where $p_K^{\mathrm{pre}}(\cdot)$ is a marginal distribution at $K$ induced by pre-trained policies. Then, by induction, during reward optimization, we show that $k \in [K]$: $x_k$ follows $p_k^{\mathrm{pre}}(\cdot)\exp(v_k(\cdot)/\alpha)/C$. Then, when $k = 0$, it would be equal to $p^{\mathrm{pre}}(\cdot)\exp(r(\cdot)/\alpha)$.

## 5. Practical Design of Algorithms

As mentioned, **RERD** is a unified sequential refinement framework that can integrate off-the-shelf approximation strategies during reward-guided denoising (Line 4 in Algorithm 1). A key practical consideration is determining which approximation methods to adopt. In this context, we present a specific version that bears similarities to evolutionary algorithms.

### 5.1. Combining Local IS and Global Resampling

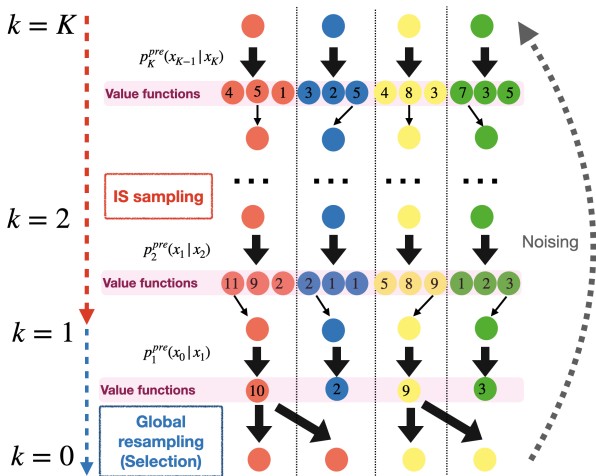

Figure 4: Visualization of Algorithm 2. A reward-guided denoising consists of two components: local value-weighted sampling for each sample (from $k = K$ to $k = 1$) and global resampling among samples in a batch at $k = 1$.

Our specific recommendation for approximating soft optimal policies during reward-guided denoising (Line 4 in Algorithm 1) is presented in Algorithm 2. Here, we adopt a strategy that does *not* require differentiable value function models, as reward feedback could often be provided in a black-box manner (e.g., molecular design). Specifically, we organically combine IS-based and SMC-based approximations. Given a batch of samples, we apply IS from $k = K$ to $k = 1$ (Line 4-6) *for each sample in the batch*, where the proposal distribution is a policy from pre-trained diffusion models. However, at the terminal step $k = 1$, we perform selection via resampling (Line 7), which is central to SMC and evolutionary algorithms. This step involves *interaction among samples in the batch*, as illustrated in Figure 4.

---

**Algorithm 2** Practical version of **RERD**

---

1: **Require**: Estimated value functions $\{\hat{v}_t\}_{t=T}^0$ (i.e., $\{r(\hat{x}_0(x_t))\}_{t=T}^0$), pre-trained diffusion models $\{p_k^{\text{pre}}\}_{k=T+1}^1$, initial designs $\{x_{0,i}^{\langle 0 \rangle}\}_{i=1}^N$ (the index $\langle \cdot \rangle$ means the number of iteration steps and the index $i \in [N]$ is an index in a batch), duplication number $L$ in IS, repetition number $S$, noise level $K$, $\alpha \in \mathbb{R}$

2: **for** $s \in [0, \cdots, S-1]$ **do**

3:     *Noising:* For each $i \in [N]$, sample $x_{K,i}^{\langle s+1 \rangle}$ from forward noising processes $q_K(\cdot \mid x_{0,i}^{\langle s \rangle})$.

4:     **for** $k \in [K-1, \cdots, 1]$ **do**

5:         IS: Sample $\forall i \in [N]$, $\{z_{k,i,l}\}_{l=1}^L \sim p_{k+1}^{\text{pre}}(\cdot \mid x_{k+1,i}^{\langle s+1 \rangle})$ and define next states from the weighted empirical distributions:

$$\forall i : x_{k,i}^{\langle s+1 \rangle} \sim \sum_{l=1}^L w_l \delta_{z_{k,i,l}}, w_l = \frac{\exp(r(\hat{x}_0(z_{k,i,l}))/\alpha)}{\sum_s \exp(r(\hat{x}_0(z_{k,i,s}))/\alpha)}.$$

6:     **end for**

7:     *Selection*: $\forall i \in [N]$, sample $x_{0,i} \sim p_1^{\text{pre}}(\cdot \mid x_{1,i}^{\langle s+1 \rangle})$ and perform resampling:

$$\{x_{0,i}^{\langle s+1 \rangle}\}_{i=1}^N \sim \sum_{i=1}^N w_i \delta_{x_{0,i}}, \ w_i = \frac{\exp(r(x_{0,i})/\alpha)}{\sum_s \exp(r(x_{0,s})/\alpha)}.$$

8: **end for**

9: **Output**: $\{x_{0,i}^{\langle S \rangle}\}_{i=1}^N$

---

This combined strategy during reward-guided denoising leverages the advantages of both IS approaches (Li et al., 2024) and SMC approaches (Wu et al., 2024). First, if we use the pure IS strategy from $k = K$ to $k = 1$, when a sample in a batch is poor, it will not be permanently discarded during the refinement process. In contrast, in Algorithm 2, the final selection step allows for the elimination of such poor samples through resampling. Second, if we use the pure SMC strategy from $k = K$ to $k = 1$, resampling is performed at every time step, which significantly reduces the diversity among samples in the batch. We apply the SMC approach only at the final step.

**Relation to evolutionary algorithm.** The above version can be viewed as a modern variant of the evolutionary algorithm, which seamlessly integrates diffusion models. An evolutionary algorithm typically consists of two steps: (a) candidate generation via mutation and crossover and (b) selection. In Algorithm 2, the step (a) corresponds to Lines 3-6, where reward-guided generation is employed, and the step corresponds to Line 7.

**Remark 1.** *When the reward feedback is differentiable, we can effectively integrate classifier guidance into the proposal distributions. For further details, see the Appendix in Li et al. (2024).*

## 5.2. Constrained Reward Optimization

We often need to include hard constraints so that generated designs fulfill certain conditions. This is especially crucial in molecular design, where we may require low-toxicity small molecules or cell-type–specific DNA sequences, as shown in Section 6.2. Here, we explore how to enable generation under such constraints. Formally, we define the constraint set as $\mathcal{C} = \{x : r_2(x) < c\}$. Given another reward $r_1(\cdot)$ to be optimized, our objective is to produce designs with high $r_1(\cdot)$ while ensuring $r_2(x) < c$.

**Naïve approaches with single-shot algorithms.** As an initial consideration, we examine how to address this problem using existing single-shot methods. A straightforward approach is to use the following reward

$$r(\cdot) = r_1(\cdot) I(r_2(\cdot) < c)$$

or use a log barrier formulation:

$$r(\cdot) = r_1(\cdot) + \log(\max(c - r_2(\cdot), c_1)),$$

where $c_1$ is a suitably small value, and then sample from $t = T$ to $t = 0$ by following approxima soft-optimal policies. However, in reality, the outputs at $t = 0$ often fail to satisfy these constraints, regardless of how the rewards are defined. This shortcoming arises because the value function models used during reward-guided denoising are not completely accurate.

**Integration into our proposal (Algorithm 2).** Now, we consider incorporating the above rewards into our framework in Algorithm 2. Here, compared to single-shot algorithms, we can often begin with feasible initial designs that satisfy the constraints $x \in \mathcal{C}$. Then, by keeping the noise level $K$ in Algorithm 2 small, we can avoid deviating substantially from these feasible regions. This gradual refinement strategy makes it easier to produce designs that fulfill hard constraints.

## 6. Experiment

We aim to evaluate the performance of the proposed method (**RERD**) across several tasks by investigating the effectiveness of refinement procedures compared to existing single-shot guidance methods in diffusion models. We begin by introducing the baselines and metrics used in our evaluation. Subsequently, we present our results in protein and DNA design. For further details and additional results, refer to Section C. The code is available at https://github.com/masa-ue/ProDifEvo-Refinement.

**Baselines and our proposal.** We compare baselines that address reward-guided generation in diffusion models with **RERD**. Note that we primarily focus on settings where reward feedback is provided in a black-box manner.

Table 1: The results for the protein design task show that our method consistently outperforms the baselines. Note that P50 and P95 represent the median and 95% quantile of the rewards for generated designs, respectively. LL denotes the (estimated) per-residue log-likelihood. Values in parentheses represent the estimated 95% standard deviation.

| Task | (a) ss-match | | | (b) cRMSD | | | (c) globularity | | | (d) symmetry | | |
|---|---|---|---|---|---|---|---|---|---|---|---|---|
| | P50 ↑ | P95 ↑ | LL ↑ | P50 ↓ | P95 ↓ | LL ↑ | P50 ↑ | P95 ↑ | LL ↑ | P50 ↑ | P95 ↑ | LL ↑ |
| SMC | 0.63 (0.04) | 0.80 | -3.28 | 8.9 (0.7) | 5.1 | **-3.58** | -2.79 (0.05) | -2.13 | -4.43 | -0.45 (0.03) | 0.21 | -3.30 |
| SVDD | 0.66 (0.02) | 0.82 | **-3.03** | 8.2 (0.4) | 4.6 | -3.59 | -2.45 (0.02) | -2.00 | -4.68 | -0.33 (0.04) | 0.36 | -3.56 |
| GA | 0.70 (0.00) | 0.95 | -3.51 | 6.3 (0.4) | 3.01 | -3.60 | -1.35 (0.02) | -1.22 | **-4.38** | 0.21 (0.04) | 0.44 | **-3.07** |
| **RERD** | **0.86** (0.01) | **0.96** | -3.13 | **1.68 (0.02)** | **0.96** | -3.51 | **-1.29** (0.02) | **-1.15** | -4.45 | **0.34** (0.01) | **0.69** | -3.08 |

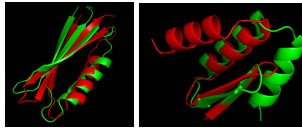
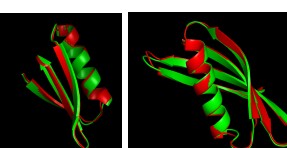
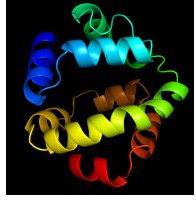
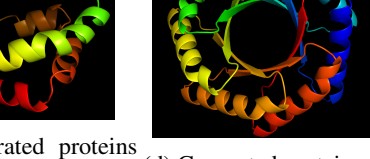
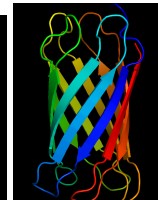

(a) Generated proteins (Green) when optimizing **ss-match** are shown. Red represents the target secondary structures. The **ss-match** score for the left figure is 0.96, while for the right figure, it is 1.0.

(b) Generated proteins (Green) from **RERD** when **cRMSD** are shown. Red represents the target backbone structures. The **cRMSD** score for the left figure is 0.42, while for the right figure, it is 0.6.

(c) Generated proteins when optimizing **globularity**.

(d) Generated proteins when optimizing **symmetry**

Figure 5: We visualize the sequences generated from **RERD** using ESMFold.

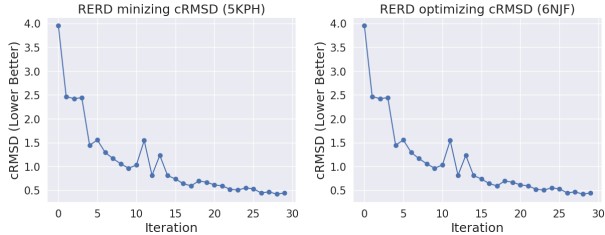

Figure 6: The refinement step from **RERD** when optimizing cRMSD in two target backbone structures is demonstrated. Recall that the first iteration corresponds to the result from SVDD. The Y-axis represents the median reward of generated samples (**Lower is better**).

- **SVDD (Li et al., 2024)**: A representative single-shot, derivative-free guidance method (without refinement).

- **SMC (Wu et al., 2024)**: Another single-shot, representative derivative-free guidance method.

- **GA**: A naïve approach for sequence design that uses pre-trained diffusion models to generate mutated designs within a standard genetic algorithm (GA) pipeline (Hie et al., 2022). To ensure a fair comparison, we allocate the same computational budget as **RERD** below.

- **RERD** in Algorithm 2 (Ours). We set $K/T = 10\%$ and $S = 50$. For initial designs, we use the results generated by SVDD in Section 6.1 and designs that satisfy the constraints in Section 6.2.

Note that we have used the same hyperparameters $\alpha, L$ across baselines (SMC, SVDD) and **RERD**.

**Metrics.** We report the top 95% quantile (**P95**) and median of rewards (**P50**) from generated designs, as these are the primary metrics to optimize. Additionally, we present the estimated per-residue log-likelihood (**LL**) using the pretrained diffusion models, which serves as a secondary metric that we aim to maintain at a moderately high value to preserve the naturalness of the designs. [1]

### 6.1. Protein Design

We begin by outlining our tasks. First, we use EvoDiff (Alamdari et al., 2023), a representative discrete diffusion model for protein sequences trained on the UniRef database, as our unconditional base model. Next, following existing representative works in protein design (Hie et al., 2022; Watson et al., 2023; Ingraham et al., 2023), we consider four reward functions related to structural properties, which take the generated sequence as input. For more details, refer to Section C.

- **ss-match**: We use Biotite (Kunzmann and Hamacher, 2018) to predict the secondary structure (ss). We then calculate the mean matching probability across all residues between the predicted and reference secondary

---

[1] We also report the diversity of generated designs. Since this metric is difficult to compare formally and secondary in the context of reward optimization, it is included in the Appendix.

Table 2: The results for the DNA design task show that our method consistently outperforms the baselines.

| Task | HepG2 | | | K562 | | | SKNSH | | |
|------|-------|---|---|------|---|---|-------|---|---|
| | P50 ↑ | P95 ↑ | LL ↑ | P50 ↓ | P95 ↓ | LL ↑ | P50 ↑ | P95 ↑ | LL ↑ |
| SMC | 1.2 (0.3) | 1.6 | -1.15 | 1.0 (0.2) | 1.4 | **-1.21** | 0.8 (0.2) | 1.0 | -1.22 |
| SVDD | 2.3 (0.2) | 2.8 | **-1.08** | 1.3 (0.3) | 1.6 | -1.26 | 1.7 (0.3) | 2.0 | **-1.21** |
| GA | 2.3 (0.4) | 2.7 | -1.21 | 2.2 (0.3) | 2.6 | -1.31 | 1.9 (0.4) | 2.5 | -1.28 |
| **RERD** | **7.9** (0.2) | **9.1** | -1.18 | **7.4** (0.2) | **8.9** | -1.25 | **5.5** (0.2) | **6.7** | -1.24 |

structures, where the target structure is represented by a sequence consisting of $a$ ($\alpha$-helices), $b$ ($\beta$-sheets), and $c$ (coils). A score of $1.0$ indicates perfect alignment.

- **cRMSD**: This is the constrained root mean square deviation against the reference backbone structure after structural alignment. Typically, $< 2\text{Å}$ indicates a highly similar structure. Note that a lower value is preferred.

- **globularity (+ pLDDT)**: It reflects how closely the structure resembles a globular shape. Additionally, we optimize **pLDDT** to improve the stability of the structure.

- **symmetry (+pLDDT, hydrophobicity)**: It indicates the symmetry of the structure in the generated sequence. Additionally, we optimize **pLDDT** and **hydrophobicity** to improve the stability of the structure.

Note that each of the above rewards is computed after estimating the corresponding structure using ESMFold (Lin et al., 2023). Besides, for both **ss-match** and **cRMSD**, we use 10 reference proteins randomly chosen from datasets in Dauparas et al. (2022) and report the mean of the results.

**Results.** We present our performance in Table 3 and visualize generated sequences in Figure 5. Overall, our algorithm (**RERD**) consistently demonstrates superior performance in terms of rewards while maintaining reasonably high likelihood. Notably, as illustrated in Figure 6, for several challenging tasks, while one-shot guidance methods such as SVDD underperforms, our approach, with refinement steps, gradually yields improved results.

### 6.2. Cell-Type-Specific Regulatory DNA Design

We begin by outlining our tasks. Here, we focus on widely studied cell-type-specific regulatory DNA designs, which are crucial for cell engineering (Taskiran et al., 2024; Su et al., 2025). Specifically, our goal is to design enhancers (i.e., DNA sequences that regulate gene expression) that exhibit high activity levels in certain cell lines while maintaining low activity in others.

Following existing works (Lal et al., 2024; Sarkar et al., 2024; Gosai et al., 2023), we construct reward functions

as follows. Using datasets from Gosai et al. (2023), which measures the enhancer activity of 700k DNA sequences (200-bp length) in human cell lines using massively parallel reporter assays (MPRAs), we trained oracles based on the Enformer architecture (Avsec et al., 2021) as rewards across three cell lines ($r_\text{H}(\cdot)$ in HepG2 cell line, $r_\text{K}(\cdot)$ in K562 cell line , and $r_\text{S}(\cdot)$ in SKNSH cell line). Then, we aim to respectively optimize the following:

$$\bar{r}_\text{H}(x) = r_\text{H}(x)\text{I}(r_\text{K}(x) < c)\text{I}(r_\text{S}(x) < c) \qquad (4)$$

where $c$ is a threshold. Here, optimizing $\bar{r}_\text{H}$ means maximizing $r_\text{H}$ while retaining $r_\text{K}, r_\text{S}$ low. Then, similarly, we define $\bar{r}_\text{K}, \bar{r}_\text{S}$ by exchanging their roles.

Here are several additional points to note. First, as discussed in Section 5.2, directly using $\bar{r}_H, \bar{r}_K, \bar{r}_S$ in practice would lead to suboptimal performance. Therefore, we use log barrier reward functions for all methods. Additionally, for **GA** and **RERD**, we initialize the designs with samples that satisfy the constraints (e.g., $\text{I}(r_\text{K}(x) < c)\text{I}(r_\text{S}(x) < c)$)). Recall that one of the advantages of our method is its ability to leverage designs from feasible regions that satisfy the constraints. Finally, we use pre-trained discrete diffusion models from Wang et al. (2024a) as the backbone unconditional diffusion models.

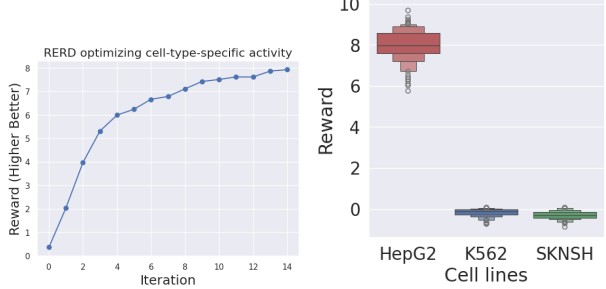

Figure 7: (Left) The refinement step from **RERD** is demonstrated. The Y-axis represents the median reward of generated samples (**Higher is better**), (Right) Generated designs from IRAO. It is seen that the activity in the target cell line HepG2 is only high.

**Results.** The results are presented in Table 2. Our methods consistently exhibit superior performance in terms of

rewards while maintaining a relatively high likelihood. Notably, while it has been reported that **SMC** and **SVDD** excel in optimizing individual rewards (e.g., $r_H$ only) in existing works such as Li et al. (2024), we have observed that they struggle with handling additional constraints. In contrast, as shown in Figure 7, **RERD** effectively handles such constraints (i.e., ensuring cell-type specificity) by gradually refining the results, starting from designs in feasible regions.

## 7. Conclusion

We introduce a new framework for inference-time reward optimization in diffusion models, utilizing an iterative evolutionary refinement process. We also provide a theoretical guarantee for the framework's effectiveness and demonstrate its superior empirical performance in protein and DNA design, surpassing existing single-shot reward-guided generation algorithms. As future work, we plan to explore its application in small-molecule design.

## Impact Statement

This paper presents a novel framework for inference-time reward optimization in diffusion models, with applications in protein and DNA design. Our work aims to improve the ability of generative models to optimize structured objectives through iterative refinement. While this research primarily contributes to technical advancements in generative modeling, it has potential implications in domains such as drug discovery and biomolecular engineering, where optimizing protein structures can accelerate scientific innovation.

We acknowledge that generative models, particularly those optimized for specific reward functions, could be misused if not carefully applied, such as in the design of harmful biological agents. However, our work is intended for scientific and therapeutic applications, and we emphasize the importance of responsible deployment and alignment with ethical guidelines in biotechnology. Overall, our contributions align with the broader goal of machine learning methodologies, and we do not foresee any immediate ethical concerns beyond those generally associated with generative models.

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

## A. Proof of Theorem 1

Here, we use induction. Hence, we prove that $x_0^{\langle 1 \rangle}$ follows $p^{(\alpha)}$.

**Distribution after noising.** First, we consider the distribution after noising. This is

$$\int q_K(x_K^{\langle 1 \rangle} \mid x_0^{\langle 0 \rangle}) p^{(\alpha)}(x_0^{\langle 0 \rangle}) dx_0^{\langle 0 \rangle}.$$

By plugging in the first assumption regarding distributions of initial designs, it is equal to

$$\int q_K(x_0^{\langle 1 \rangle} \mid x_K^{\langle 1 \rangle}) q_K(x_K^{\langle 1 \rangle}) \exp(r(x_0^{\langle 0 \rangle})/\alpha) dx_0^{\langle 0 \rangle}. \tag{5}$$

Recalling this definition of soft value functions:

$$\exp(v_K(\cdot)/\alpha) = \mathbb{E}_{p^{\mathrm{pre}}(x_0|x_K)}[\exp(r(x_0)/\alpha) \mid x_K]$$

and the assumption (b) ($q_0(x_0|x_K) = p^{\mathrm{pre}}(x_0|x_K)$ and $q_K(\cdot) = p_K^{\mathrm{pre}}(\cdot)$ ), the term (5) is equal to

$$p_K^{\mathrm{pre}}(\cdot) \exp(v_K(\cdot)/\alpha))/C.$$

**Distribution after reward-guided denoising.** Now, we consider the distribution of $x_0^{\langle 1 \rangle}$:

$$1/C \int \left\{ \prod_{k=K}^{1} p_k^\star(x_{k-1} \mid x_k) \right\} p_K^{\mathrm{pre}}(x_K) \exp(v_K(x_K)/\alpha)) d(x_0, \cdots, x_K).$$

With some simple algebra, this is equal to

$$1/C \int \left\{ \prod_{k=K-1}^{1} p_k^\star(x_{k-1} \mid x_k) \right\} \times \frac{p_K^{\mathrm{pre}}(x_{K-1}|x_K) \exp(v_{K-1}(x_{K-1})/\alpha))}{\exp(v_K(x_K)/\alpha))} \times p_K^{\mathrm{pre}}(x_K) \exp(v_K(x_K)/\alpha)) d(x_0, \cdots, x_K)$$

$$= 1/C \int \left\{ \prod_{k=K-1}^{1} p_k^\star(x_{k-1} \mid x_k) \right\} \times p_K^{\mathrm{pre}}(x_{K-1}|x_K) p_K^{\mathrm{pre}}(x_K) \exp(v_{K-1}(x_{K-1})/\alpha)) d(x_0, \cdots, x_K)$$

$$= 1/C \int \left\{ \prod_{k=K-1}^{1} p_k^\star(x_{k-1} \mid x_k) \right\} p_{K-1}^{\mathrm{pre}}(x_{K-1}) \exp(v_{K-1}(x_{K-1})/\alpha)) d(x_0, \cdots, x_{K-1}).$$

Repeating this argument from $k = K - 1$ to $k = 0$, the above is equal to

$$p_0^{\mathrm{pre}}(\cdot) \exp(r(\cdot)/\alpha)/C.$$

This concludes the statement.

## B. Additional Details for Protein Design

In this section, we have added further details on experimental settings and results.

### B.1. Details on Baselines

- **RERD** (Algorithm 2): We have used parameters $L = 20, N = 10, S = 30$ in general. For the importance sampling step, we have used $\alpha = 0.0$, and for the selection step, we have used $\alpha = 0.2$.

- **SVDD**: We set the tree width $L = 20, \alpha = 0.0$.

- **SMC**: In SMC, we set $\alpha = 0.05$ because if we choose $\alpha = 0.00$, it just gives a single sample every time step. Refer to Appendix B in Li et al. (2024).

- **GA**: Here, compared to Algorithm 2, we have changed the mutation part (Line 3-7) with just sampling from pre-trained diffusing models without any reward-guided generation. To have a fair comparison with **RERD**, we increase the repetition number $S$ so that the computational budget is roughly the same as our proposal.

## B.2. Details on Reward Functions

**Globularity.**  Globularity refers to the degree to which a protein adopts a compact and nearly spherical three-dimension structure (Pace and Hermans, 1975).It is defined based on the spatial arrangement of backbone atomic coordinates, where the variance of the distances between those coordinates and the centroid is minimized, leading to a highly compact structure. Here, we set the protein length 150.

Globular proteins are characterized by their structure stability and water solubility, differing from fibrous or membrane proteins. The compact conformation helps proteins to maintain proper protein folding and reduce the risk of aggregation.

**Symmetry.**  Protein symmetry refers to the degree to which protein subunits are arranged in a repeating structure pattern (Goodsell and Olson, 2000; Lisanza et al., 2024; Hie et al., 2022). Here we focus on the rotational symmetry of a single chain, which is defined by the spatial organization of subunit centroids. Specifically, we try to minimize the variances of the distances between adjacent centroids to achieve a more uniform and balanced arrangement. Here, we set the protein length to be 150 to 240.

Symmetric proteins can bring multiple functional sites into close proximity, facilitating interactions and supporting the formation of large proteins with optimized biological functions.

**Hydrophobicity.**  Hydrophobicity refers to the degree to which a protein repels water, primarily defined by the distribution of hydrophobic amino acids within the structure, namely, Valine, Isoleucine, Leucine, Phenylalanine, Methionine and Tryptophan (Chandler, 2002). Hydrophobicity is optimized by minimizing the average Solvent Accessible Surface Area (SASA) of the hydrophobic residues above, thus reducing their exposure to the surrounding solvent. Hydrophobicity enhances the protein structural stability, especially in the polar solvents such as water, facilitates the protein-protein interactions by prompting binding at the hydrophobic surfaces, and drives the proper protein folding by guiding the hydrophobic residues to the protein core.

**pLDDT.**  pLDDT (predicted Local Distance Difference Test) is a confidence score used to evaluate the reliability of the local structure in predicted proteins. It is defined by the confidence of model predictions, assigning a confidence value to each residues. A higher pLDDT score indicates greater model confidence and suggests increased structural stability. To optimize the whole protein structure, we try to maximize the average pLDDT across the whole sequence as predicted by ESMFold (Lin et al., 2023).

## B.3. Additional Results

**More metric (diversity, pLDDT, and pTM).**  We have included additional metrics in Table 3.

- Generally, higher pLDDT and pTM values indicate more accurate structure predictions at the local residue and the global structure, respectively. However, in the context of de novo protein design, a low pLDDT does not necessarily imply poor performance (Verkuil et al., 2022). In the globularity task, it is expected that the generated protein is more novel protein.

- We define diversity as 1 - the mean pairwise distance (normalized by length), where the distance is measured using the Levenshtein distance. While diversity can be an important metric to evaluate the performance of pre-trained generative models, in the context of reward optimization, this metric may be secondary. It is shown that generated sequences from **RERD** are reasonably diverse enough without collapsing to single samples.

Table 3: Additional metrics for experiments in protein design. We have reported the median of pLDDT, pTM, and diversity of generated proteins.

| Task | (a) ss-match | | | (b) cRMSD | | | (c) globularity | | | (d) symmetric | | |
| --- | --- | --- | --- | --- | --- | --- | --- | --- | --- | --- | --- | --- |
| | pLDDT | pTM | diversity | pLDDT | pTM | diversity | pLDDT | pTM | diversity | pLDDT | pTM | diversity |
| **RERD** | 0.75 | 0.69 | 0.28 | 0.76 | 0.71 | 0.14 | 0.41 | 0.29 | 0.56 | 0.82 | 0.79 | 0.49 |

**Recovery rate when optimizing cRMSD.** By optimizing cRMSD, we can tackle the inverse folding task. While we have not extensively investigated the performance in terms of recovery rates, we present the observed recovery rates for several proteins as a reference when using **RERD**. Although it does not match the performance of state-of-the-art conditional generative models specifically trained for this task, such as ProteinMPNN (Dauparas et al., 2022), our algorithm, which combines *unconditional* diffusion models with reward models at *test-time*, demonstrates competitive performance.

Table 4: Recovery rates when optimizing cRMSD

| Proteins | 5KPH | 6NJF | EHEE _rd1_0101 | EA:run2 _0325_0005 |
|---|---|---|---|---|
| **RERD** | 0.26 | 0.31 | 0.28 | 0.30 |
| ProteinMPNN | 0.41 | 0.53 | 0.35 | 0.38 |

**More generated proteins.** We have visualized more generated proteins in Figure 8.

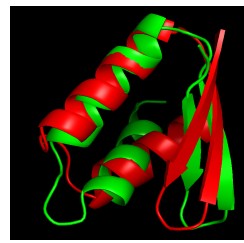
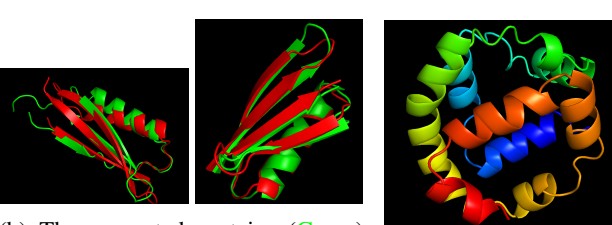
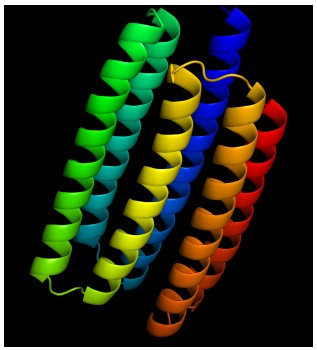

(a) The generated proteins (Green) when optimizing **ss-match** are shown. Red represents the target secondary structures. The **ss-match** score is 1.0 here.

(b) The generated proteins (Green) when optimizing **cRMSD** are shown. Red represents the target secondary structures.

(c) The generated proteins when optimizing **globularity** are shown.

(d) The generated proteins when optimizing **symmetry** are shown.

Figure 8: More generated protein from **RERD**.

# C. Additional Details for DNA Design

**Pre-trained models.** We use the pre-trained diffusion model trained in Wang et al. (2024b). The code and its performance are available in their paper. Here, we use the discrete diffusion model proposed in (Sahoo et al., 2024) using the same CNN architecture as in (Stark et al., 2024) and a linear noise schedule.

**Reward oracles.** We use the exact oracle used in Wang et al. (2024b). Again, the code and its performance are available in their paper. Here, we use the Enformer architecture (Avsec et al., 2021) initialized with its pretrained weights. We use the data splitting based on chromosome following standard practice (Lal et al., 2024).

**Hyperparameters in baselines and RERD.** We set $S = 15, \alpha = 0.0, L = 20$.

**Diversity.** We calculate the diversity as in the protein design task. It is 0.47 in HepG2, 0.49 in K562, and 0.53 in SKNSH. It is shown that generated sequences are reasonably diverse enough.

