# OpenReview forum: "Reward-Guided Iterative Refinement in Diffusion Models at Test-Time with Applications to Protein and DNA Design"
_ICML.cc/2025/Conference — ICML 2025 poster_

### Official Review · Reviewer_HSgw · 2025-03-13

**Overall Recommendation:** 3

**Summary:**

The paper introduces 'Reward-Guided Evolutionary Refinement in Diffusion models (RERD)', a framework for optimizing reward functions during inference time in diffusion models. RERD employs an iterative refinement process consisting of two key steps per iteration: noising and reward-guided denoising. This approach enhances downstream reward functions while preserving the naturalness of generated designs. The framework is backed by a theoretical guarantee and demonstrates superior performance in protein and DNA design tasks compared to single-shot reward-guided generation methods.

**Claims And Evidence:**

The claims are not fully substantiated by clear and compelling evidence. The author suggests setting K/T to a low value; however, there is no ablation study on the noise scale K. Without such an analysis, it is difficult to assess the model’s performance in terms of reward estimation and computational cost across different K values. Additionally, while the paper asserts broad applicability to all diffusion models, the proposed method is only implemented on discrete models, raising questions about its generalizability.

**Essential References Not Discussed:**

No

**Experimental Designs Or Analyses:**

Yes

**Methods And Evaluation Criteria:**

Yes, the proposed method aims to overcome the limitations of single-shot approaches in optimizing complex rewards and managing hard constraints, which is relevant to the protein design task.

**Other Comments Or Suggestions:**

None

**Other Strengths And Weaknesses:**

Weaknesses:
1. Limited Impact Due to Model Choice – The use of the relatively less popular EvoDiff model for protein design may restrict the broader influence and adoption of the work within the field.
2. Limited Novelty – The proposed method shares similarities with [1], which also employs a resampling-based correction approach for diffusion models, reducing the novelty of the contribution.

[1] Liu, Yujian, et al. "Correcting diffusion generation through resampling." Proceedings of the IEEE/CVF Conference on Computer Vision and Pattern Recognition. 2024.

**Questions For Authors:**

None

**Relation To Broader Scientific Literature:**

The paper is related to finetuning diffusion models with guidance and inference-time scaling for diffusion.
The paper is also related to applying RL for protein design.

**Theoretical Claims:**

Yes

---

> ### Author Rebuttal · Authors · 2025-04-01
>
> Thank you for your constructive suggestions and insightful comments! Following a reviewer's suggestion, we added (1) more ablation studies and (2) additional experiments for image generation with Stable Diffusion and MaskGiT
>
> > Ablation studies on the noising faction (K)
>
> Thank you for the thoughtful suggestions regarding the ablations. In response, we have performed additional ablation studies by varying key hyperparameters. To provide a quick yet informative signal, we focused on the ss-match and cRMSD tasks. Here is a  [link](https://anonymous.4open.science/r/hidden-1886) to figures describing experimental results.
> We plan to extend these studies to other tasks in the final version.
>
> - We added an ablation study varying K (**Figure 1,2** in the link) by fixing the computational budget for evaluating reward models. The results show a strong performance when K/T=10% or 20%. Generally, a large K/T reduces the benefit of refinement, while a very small K/T limits the opportunity for reward-guided decoding.
>
> - We also performed an ablation on L, the number of repetitions for importance sampling (**Figure 3,4** in the link). As expected, performance improves with a larger L due to the increased computational budget.
>
> > The proposed method is only implemented on discrete models
>
> That's a great point! We’ve focused on discrete diffusion models because they tend to have a greater impact in the protein design domain. That said, our method can integrated with continuous diffusion models as well. To verify this, we have implemented our method when **we set Stable diffusion as pre-trained continuous diffusion models** and compressibility (the negative file size in kilobytes (kb) of the image after JPEG compression) as reward models (Figure 5 in the link). Following our experiment section, we have tried two scenarios where we set K/T to be 10% and 20%.  This figure also highlights the effectiveness of iterative refinement in continuous diffusion models, as we showed in our protein design scenarios (Figure 6 in our original draft).  We will incorporate them in the final version.
>
> > Limited novelty over [1]
>
> Thank you for pointing out this work—we will certainly include a citation in the revised version. From our understanding, the paper introduces an SMC-based approach similar to other related methods we have cited (e.g., Wu et al., 2024; Dou and Song, 2024). However, it appears to follow a more single-shot sampling strategy. While the restart sampler component may share a similar spirit, **Our main contribution—an iterative refinement procedure tailored for reward optimization, supported by both theoretical and empirical evidence**—differs substantially in both methodology and intent. We will make this distinction clearer in the final version.
>
> > Limited Impact Due to Model Choice (EvoDiff is less popular)
>
> 1. To the best of our knowledge, EvoDiff is widely recognized as a representative discrete diffusion model in the protein design domain, as noted in recent reviews (e.g., Winnifrith et al.). While other pre-trained diffusion models, such as DPLM and ESM-3, are also potential candidates, incorporating them into our framework would be relatively straightforward. We are be happy to include additional results if the reviewer has specific protein diffusion models in mind.
>
> Winnifrith, Adam, Carlos Outeiral, and Brian L. Hie. "Generative artificial intelligence for de novo protein design." Current Opinion in Structural Biology 86 (2024): 102794.
>
> 2. To further address the reviewers’ concerns, **we conducted additional experiments on image generation tasks using a different discrete diffusion model, implemented on top of the [MaskGiT](https://masked-generative-image-transformer.github.io/) codebase**.  Here, the setup closely resembles that of EvoDiff. We set the duplication number to L=20, and in each iteration, we remask a 10% square region of the entire image. Experiments are conducted across 32 image categories. As shown in **Figure 6 (linked above)**, compressibility consistently improves over iterations, demonstrating the practical effectiveness of RERD. One point of clarification: the compressibility here appears higher than that observed with Stable Diffusion earlier. This is primarily because MaskGiT operates in compressed sequence spaces, making it easier to optimize.
>
> These results indicate that our method performs robustly across various pre-trained models. We will include more comprehensive quantitative results in the final version, and of course, we would be happy to provide further clarifications during the rebuttal process.

---

> > ### Comment · Reviewer_HSgw · 2025-04-04
> >
> > The authors have addressed most of my concerns and I have raised my score to weak accept.

---

> > > ### Author Response · Authors · 2025-04-09
> > >
> > > Thank you for your support and valuable suggestions again! We will carefully incorporate them into the final version.

---

### Official Review · Reviewer_t8tW · 2025-03-14

**Overall Recommendation:** 4

**Summary:**

The authors introduce a novel inference-time framework for the iterative refinement and reward optimization of diffusion models. Their proposed method, Reward-Guided Evolutionary Refinement in Diffusion models (RERD), is based on the iterative refinement of generation with reward-guides denoising, and provide theoretical support for their method. The authors demonstrate the use case of RERD on masked diffusion models for the tasks of protein and biological sequence design. Through a set of reasonably thorough experiments, they show that their method yields improved performance relative to counterpart baselines.

**Claims And Evidence:**

In general, the claims of the paper are supported by empirical evidence and theoretical results. One item I would like to point out:

- On lines 70-72 (right): "our work is the first attempt to study iterative refinement in diffusion models". I am not entirely certain this claim is true, but I could be wrong. Please take a look at my comment in the "Essential References Not Discussed" section of the review.

**Essential References Not Discussed:**

I think one related reference was missed [Domingo-Enrich et al. 2024] on the topic of "Guidance (a.k.a. test-time reward optimization) in diffusion models." To add, I encourage the authors to denote the differences between their proposed method and that of [Domingo-Enrich et al. 2024] for the refinement of diffusion models, or consider it as a baseline.

Otherwise, and to the best of my knowledge, all relevant related works are discussed.

- Domingo-Enrich, Carles, et al. "Adjoint matching: Fine-tuning flow and diffusion generative models with memoryless stochastic optimal control." arXiv preprint arXiv:2409.08861 (2024).

**Experimental Designs Or Analyses:**

The authors consider a thorough set of empirical experiments for the tasks of protein design and cell-type specific sequence design to evaluate and validate their proposed method. In general, the experiments which the authors conduct in this work appear sound and valid.

**Methods And Evaluation Criteria:**

The authors evaluate their proposed method with a diverse set of metrics and on a diverse set of tasks/settings.

**Other Comments Or Suggestions:**

N/A

**Other Strengths And Weaknesses:**

In general, I believe this is a well-written and easy-to-follow paper which showcases some convincing experimental results while also providing solid theoretical contributions to back the proposed approach.

I did not find any obvious weaknesses.

**Questions For Authors:**

1. From algorithm 1 and Figure 3, does this mean you run the inference process of the diffusion model $S-1$ times? i.e. if the diffusion models uses $1000$ inference steps, this means RERD needs $100 * (S - 1)$ steps to generate a sample?

**Relation To Broader Scientific Literature:**

This paper tackles problems of two broader areas: (1) controllable generation via diffusion models, and (2) protein and sequence design. Both are active fields, where addressing items (1) and (2) would have significant impact in the respective areas. I believe this work makes a sound contribution to both fields.

**Theoretical Claims:**

Theorem 1 is the primary theoretical claim and is supported by a proof, which seems correct.

---

> ### Author Rebuttal · Authors · 2025-04-01
>
> We sincerely appreciate the positive feedback. Below are our responses to your questions:
>
> Q: Do we need 100∗(S−1) steps?
>
> You're absolutely right. When setting K/T=10%, and T=1000, we would indeed require 100 steps. However, this component can be adjusted in practice by reducing T or K, which offers flexibility depending on computational constraints. We will clarify this point in the revised version.
>
> Q: Relation to Domingo-Enrich, Carles, et al.
>
> Thank you for bringing this work to our attention. We will cite it in the final version. It appears that the focus of this paper is more on the fine-tuning of diffusion models, whereas our work emphasizes inference-time reward optimization. Following prior work such as DPS, SVDD, and SMC-based methods, we have primarily focused on comparison between inference-time techniques, which we view as complementary/orthogonal to fine-tuning approaches. That said, we agree that a more detailed discussion would be valuable and will include a comparison with fine-tuning methods in the revision.

---

### Official Review · Reviewer_xE3K · 2025-03-14

**Overall Recommendation:** 4

**Summary:**

The paper presents a novel framework for inference time reward optimization in diffusion models, introducing an iterative refinement approach that alternates between noising and reward guided denoising steps. This method departs from conventional single shot reward optimization, aiming to iteratively refine generated samples, allowing for the correction of errors and more effective optimization of complex reward functions. The authors provide a theoretical guarantee showing that their framework samples from a distribution proportional to the pretrained model distribution, weighted by the exponentiated reward function. The method is evaluated empirically on protein and DNA sequence design, demonstrating improvements over baseline approaches in optimizing structural properties of proteins and regulatory activity of DNA sequences while maintaining sample quality. The results suggest that this iterative approach is particularly useful for handling hard constraints, which is relevant in biological design tasks where feasibility constraints are often strict.

**Claims And Evidence:**

The key argument is that single shot reward guided denoising methods are limited in their ability to optimize complex reward functions due to approximation errors in estimating value functions, particularly at highly noised states. According to the authors, an iterative refinement process that progressively applies reward optimization (i.e. RERD, the proposed approach) corrects errors more gradually, can correct suboptimal decisions in earlier steps and leads to superior performance on complex reward functions that involve structural constraints.

The authors provide a theoretical justification demonstrating that RERD samples from a distribution proportional to the pre-trained model distribution weighted by the exponentiated reward function, ensuring alignment with the target reward optimized distribution. This theoretical result is derived under the assumption that the noising and denoising processes used in the iterative refinement step match the pre-trained diffusion model's forward and reverse processes. While this assumption is reasonable given the training procedure of diffusion models, the practical effectiveness of this theoretical guarantee depends on the accuracy of approximating the soft optimal policy at each step, which is not explicitly analyzed in the theoretical section.

Experimental results on protein design and regulatory DNA design tasks are presented as empirical evidence. The results consistently show that the proposed method outperforms baselines such as single shot guidance and genetic algorithms in terms of reward maximization while maintaining reasonable likelihood scores.

The paper does not include real world experimental validation beyond simulation and computational evaluation, which may limit the external validity of the claims, but within the scope of computational biomolecular design, the evidence presented is convincing.
Furthermore, evaluation is not extended to other domains where reward guided generation might be relevant. This is perhaps a limitation since the approach could be generally applicable to several domains, and evidence of this would strengthen this submission.

**Essential References Not Discussed:**

Arguably [1].

[1] https://www.biorxiv.org/content/10.1101/2024.02.01.578352v1

**Experimental Designs Or Analyses:**

The experimental design seems well structured and backs up the theoretical claims.The experiments aim to assess whether the iterative refinement process leads to superior reward optimization while maintaining biologically plausible sequences. The evaluation setup involves a combination of benchmark datasets, pretrained diffusion models, and specific reward functions tailored to each task.

For protein sequence design the authors use EvoDiff (a discrete diffusion model trained on UniRef) as the base generative model, and compute the reward functions based on structural predictions from ESMFold.
For enhancer design, the authors use a pretrained discrete diffusion model and construct reward functions using enhancer activity predictors trained on large scale datasets from [1] (with consist of measurements of enhancer activity on several DNA sequences) which they use to train predictive models based on the well known Enformer architecture. The DNA design tasks involve generating sequences that maximize enhancer activity in a target cell line while suppressing it in others, ensuring specificity to a particular cell type.

While indeed, evidently, this study relies on multiple models in the loop, this is a well known and often followed approach from various related works in the literature and often aligns with good practices in computational biology. It is still to note that these introduce an additional layer of approximation, crucially at the evaluation step.
Additional remarks: there doesn't seem to be much of a discussion of how the reference proteins were selected. Ditto for the DNA design task where the sequences are initialized from a pretrained diffusion model but their diversity is not analyzed. Also, while RERD is present as a unified framework, as mentioned previously I am also wondering about the impact of each individual component (noising, reward guided denoising, importance sampling and evolutionary resampling). The impact of each component is never examined in isolation.


[1] https://www.nature.com/articles/s41586-024-08070-z

**Methods And Evaluation Criteria:**

The proposed method RERD iteratively introduces noise to partially perturb samples before applying reward guided denoising. The denoising step uses importance sampling and a final selection mechanism inspired by evolutionary algorithms to refine samples towards high reward solutions. The motivation behind this approach is that errors introduced during reward optimization due to inaccuracies in value function approximations can be corrected over multiple iterations. The theoretical framework shows that under idealized conditions, the final samples produced by the iterative refinement process follow a distribution proportional to the pretrained diffusion model's prior distribution, reweighted by an exponentiated reward function. This ensures that the algorithm maintains a principled probabilistic framework while still allowing for effective reward optimization.

Evaluation focuses on benchmark tasks in protein and DNA sequence design with reward functions measuring structural properties and regulatory activity. These tasks have become commonplace downstream tasks to evaluate protein/biological sequence models and so are appropriate in this particular context. In particular, on protein design, secondary structure matching, backbone root mean square deviation, globularity and symmetry are used as reward metrics. All protein sequences are structurally evaluated using ESMFold. In DNA design, the task is to generate enhancer sequences that maximise activity in a specific cell type while minimizing activity in others, with reward functions designed using pretrained sequence based predictors trained on large scale enhancer activity datasets, and  evaluation metrics being 50th and 95th percentile of predicted activity scores. The baseline methods for comparison are SVDD, SMC and a genetic algorithm which applies mutations to pretrained diffusion model samples.

RERD is shown to outperform the baseline consistently (in terms of reward) while maintaining likelihood values comparable to the original diffusion model. In my opinion, the model is fairly and rigorously evaluated, and the evaluation criteria are accurate to the problem setting as it captures both reward minimisation maximization and sequence naturalness. One remark is that there is no ablation of individual components (e.g. the evolutionary resampling step).

**Other Comments Or Suggestions:**

not applicable

**Other Strengths And Weaknesses:**

Beyond what was mentioned already, the paper seems well presented and the presented framework is relatively novel work.

**Questions For Authors:**

not applicable

**Relation To Broader Scientific Literature:**

Several key contributions in this field are cited. [1] provides an in depth guide on inference time guidance methods for optimizing reward functions in diffusion models, emphasizing the need for aligning generated samples with desired metrics without retraining the model. This paper builds upon this foundation by proposing an iterative refinement process, moving beyond single shot generation. [2] explores finetuning discrete diffusion models using reinforcement learning to optimize specific reward functions, particularly in biological sequence generation. The current study is in many ways quite similar but the emphasis is placed on test time optimization.


[1] https://arxiv.org/abs/2501.09685
[2] https://arxiv.org/abs/2410.13643

**Theoretical Claims:**

The core claim is that the iterative refinement process samples from a distribution proportional to the pretrained diffusion model prior, weighted by an exponentiated reward function. This is established in Theorem 1, which asserts that under two main assumptions the final output of RERD follows the desired target distribution. These assumptions are that the initial samples follow the reward weighted distribution and that the noising process matches the forward process of the pretrained diffusion model. This claim is meant to provide a theoretical guarantee that RERD does not diverge arbitrarily from the pretrained model's learned distribution, ensuring that the generated samples remain plausible while optimizing the reward. The proof is structured as an induction argument over the iterative refinement steps, showing that if the distribution holds at step K, then applying reward guided denoising preserved this form until reaching the final step at which point the distribution matches the desired target form.
Just for my own understanding, has it been considered whether slight mismatches between the noising process and the learned forward diffusion model alter the final distribution? Also, is there a possibility that the refinement process oscillates between suboptimal solutions? Perhaps deriving a bound on the variance of samples over multiple iterations could be useful.

---

> ### Author Rebuttal · Authors · 2025-04-01
>
> Thank you very much for the positive and very detailed feedback! Below we address the key questions and comments you raised:
>
> > Q. Abolition study
>
> Thank you for the thoughtful suggestions regarding the ablations. In response, we have conducted additional ablation studies by varying key hyperparameters. To provide a quick yet informative signal, we focused on the ss-match and cRMSD tasks. Here is a link to figures describing experimental results. [link](https://anonymous.4open.science/r/hidden-1886)
> We plan to extend these studies to other tasks in the final version.
>
> - We added an ablation study varying K in Algorithm 2 (**Figure 1,2** in the link). The results show strong performance when K/T=10%  or K/T=20% This result is expected: a large K reduces the benefit of refinement, while a very small K limits the opportunity for reward-guided decoding.
>
> - We also performed an ablation on L, the number of repetitions for importance sampling (**Figure 3,4** in the link). As expected, performance improves with a larger L, likely due to the increased computational budget and exploration.
>
> > Q. Has it been considered whether slight mismatches between the noising process and the learned forward diffusion model alter the final distribution?
>
> This is an excellent point. We agree that optimization and sampling-time errors can differ in practice. While a rigorous analysis is challenging, one possible direction is to assume the mismatch is bounded in total variation distance by $\epsilon$, and then analyze how this propagates to the final distribution. We will explore this idea further and aim to incorporate a discussion in the final version.
>
> > Q: Also, is there a possibility that the refinement process oscillates between suboptimal solutions?
>
> Yes, it oscillates. But, in general, it tends to optimize a stable manner, as shown in Figure 6.
>
> > Q. How are reference proteins selected?
>
> We follow the protocol introduced by Hie et al. (2022). We will make this more explicit in the revised version.
>
> > Q. Related works.
>
> Thank you for the suggestions. We will add citations to the relevant works in the final version.

---

> > ### Comment · Reviewer_xE3K · 2025-04-09
> >
> > I thank the authors for adding clarity in response to my review. I am happy for this work to appear at ICML and will update my score to accept.

---

### Decision · Program_Chairs · 2025-05-01

**Decision:**

Accept (poster)

**Comment:**

This work introduces RERD, an iterative refinement approach for reward optimization in diffusion models which alternates between noising and reward guided denoising steps. This extends the single shot reward optimization and sequentially refines samples, increasing the ability to optimize complex reward functions. The theoretical claims are backed-up and illustrated on protein and DNA sequence design.

  All reviewers acknowledge the relevance and interest of the method, stressing the clear presentation and novelty of the framework. Some minor improvements are suggested, so I recommend acceptance as long as they are incorporated in the revised version of the paper.